# Identification and Analysis of Compound Profiles of Sinisan Based on ‘Individual Herb, Herb-Pair, Herbal Formula’ before and after Processing Using UHPLC-Q-TOF/MS Coupled with Multiple Statistical Strategy

**DOI:** 10.3390/molecules23123128

**Published:** 2018-11-29

**Authors:** Jia Zhou, Hao Cai, Sicong Tu, Yu Duan, Ke Pei, Yangyang Xu, Jing Liu, Minjie Niu, Yating Zhang, Lin Shen, Qigang Zhou

**Affiliations:** 1School of Pharmacy, Nanjing University of Chinese Medicine, Nanjing 210023, China; zhoujia19931005@126.com (J.Z.); duanyu1681@sina.com (Y.D.); yangyangxu92@126.com (Y.X.); 15951921665@163.com (J.L.); someonearis@163.com (M.N.); zhangyatingzyt@126.com (Y.Z.); sonesunfany@sina.cn (L.S.); 2Engineering Center of State Ministry of Education for Standardization of Chinese Medicine Processing, Nanjing University of Chinese Medicine, Nanjing 210023, China; 3Medical Sciences Division, University of Oxford, Oxford OX3 7BN, UK; Sicong.tu@sydney.edu.au; 4Sydney Medical School, The University of Sydney, Sydney, NSW 2006, Australia; 5Institute of Pharmaceutical and Food Engineering, Shanxi University of Traditional Chinese Medicine, Taiyuan 030024, China; peike_pk@126.com; 6School of Pharmacy, Nanjing Medical University, Nanjing 210026, China; qigangzhou@njmu.edu.cn

**Keywords:** chemical constituent profiles of Sinisan, chinese medicine processing, chinese medicinal formula compatibility

## Abstract

Sinisan has been widely used to treat depression. However, its pharmacologically-effective constituents are largely unknown, and the pharmacological effects and clinical efficacies of Sinisan-containing processed medicinal herbs may change. To address these important issues, we developed an ultra-high performance liquid chromatography coupled with electrospray ionization tandem quadrupole-time-of-flight mass spectrometry (UHPLC-Q-TOF/MS) method coupled with multiple statistical strategies to analyze the compound profiles of Sinisan, including individual herb, herb-pair, and complicated Chinese medicinal formula. As a result, 122 different constituents from individual herb, herb-pair, and complicated Chinese medicinal formula were identified totally. Through the comparison of three progressive levels, it suggests that processing herbal medicine and/or altering medicinal formula compatibility could change herbal chemical constituents, resulting in different pharmacological effects. This is also the first report that saikosaponin h/i and saikosaponin g have been identified in Sinisan.

## 1. Introduction

Chinese medicine processing and Chinese medicinal formula compatibility are two outstanding characteristics in the clinical applications of Chinese medicine. However, current studies often focus on the compatibility mechanism or processing mechanism alone without combining them together organically, and reports discussing Chinese medicine processing mechanisms in Chinese medicinal formulae have been rarely involved. Therefore, the selection of the processed products of Chinese herbal medicines contained in Chinese medicinal formulae has only to rely on the experiences of clinicians without sufficient basis of scientific theories.

Sinisan (SNS), an ancient well-known Chinese medicinal formula consisting of four Chinese herbal medicines—Bupleuri Radix (BR), Paeoniae Radix Alba (PRA), Aurantii Fructus Immaturus (AFI), and Glycyrrhizae Radix et Rhizoma Praeparata Cum Melle (GRM), has been regarded as an effective anti-depression prescription according to the traditional Chinese medicine (TCM) theories of six channels and depression. SNS was initially described by Zhongjing Zhang in ‘Treatise on Febrile Diseases’ as a traditional Chinese herbal formula to cure mental disorders. It has been widely used for thousands of years, and even today it is still the fundamental and essential prescription for the treatment of depression [1,2]. However, the application of processed BR and processed PRA contained in SNS is quite controversial, which is necessary to improve our understanding whether the processing procedures has changed any chemical constituents of the herbal medicine.

At present, ultra-high performance liquid chromatography coupled with electrospray ionization tandem quadrupole/time of flight mass spectrometry (UHPLC-Q-TOF/MS) is a powerful tool for the analysis of complex samples in TCM and possesses high resolution, efficiency, and sensitivity to obtain accurate mass information [3,4,5,6]. Multivariate statistical analysis based on all the available chemical information has made the identification of potential chemical markers possible. In this report, we successfully developed an UHPLC-Q-TOF/MS method coupled with multiple statistical strategy to analyze the compound profiles of SNS.

Based on the theory of TCM, processing with vinegar can enhance the effects of coursing liver and resolving depression [7]. A previous report has suggested that vinegar-processed BR (VPBR) is more effective in the treatment of liver disorders, including hepatitis, cirrhosis, and liver cancer [8]. In this study, we creatively analyzed the compound profiles of individual herb, herb-pair, and complicated Chinese herbal formula according to their representative herbal medicine: BR, PRA, BR-PRA herb-pair, and SNS, respectively, and also systematically compared the changes of chemical constituents of BR, PRA, BR-PRA herb-pair, and SNS before and after processing to reveal the scientific connotation of processing and formula compatibility. We are looking forward to seeking out the common mechanism of processing and formula compatibility of Chinese herbal medicine in order to provide scientific theory for safe clinical applications of Chinese medicine and rational herbal medicine processing in Chinese medicinal formula.

## 2. Results and Discussion

### 2.1. Identification of Compounds

According to the previous reports [9,10,11,12], saponins, terpenoids, and flavones are the main chemical components in BR (VPBR), PRA (VPPRA), AFI, and GRM. These components easily lose a proton under mass spectrum detection, resulting in a better mass response in negative ion mode than in positive one. As shown in Table 1, 101 compounds were identified in negative ion mode and 21 compounds were identified in positive ion mode [13,14,15]. The typical total ion chromatograms (TICs) of BR, VPBR, PRA, vinegar-processed PRA (VPPRA), BR-PRA herb-pair, VPBR-VPPRA herb-pair, SNS, and SNS-containing VPBR and VPPRA in both ion modes are shown in Figure 1.

### 2.2. Multivariate Data Analysis

Using MarkerView^TM^ 1.2.1 data handling software, multivariate data analysis were completed. The principal component analysis (PCA) score plot in negative and positive ion modes were shown in Figure 2. The results showed that all crude and processed samples including individual herb, herb-pair, and complicated Chinese medicinal formula were successfully classified into two categories in both positive and negative ion modes.

### 2.3. Compounds Changed after Processing and Formula Compatibility

The variations of components (*p* < 0.05) in the individual herb, herb-pair, and complicated Chinese herbal formula before and after processing were shown in Table 2 and Table 3. For BR, 22 peaks were shown significant differences after processing. Comparing with BR, the intensity of seven peaks increased in VPBR; the other 15 peaks declined in VPBR. Taking compatibility into consideration, it was interesting to find that 14 peaks contributing to differentiate crude and processed individual herbs disappeared in herb-pair, while three new peaks (isorhamnetin-3-rutinoside, HOSSd, 2′′-*O*-AcetylSSd) appeared. Additionally, prosaikogenin f decreased in individual herb but increased in herb-pair. Compatibility may be responsible for these changes. On the contrary, adonltol, SSh, SSi, SSg, SSb_1_, 3′′-*O*-AcetylSSa, and SSd all showed the same trend after processing of BR in the individual herb and herb-pair. Thus, it was hard to distinguish that the seven components were affected by processing, compatibility, or even their combination. Taking into further account the formula compatibility effect of AFI and GRM, eight peaks showing significant differences in herb-pair vanished in the formula, however seven new peaks (isorhamnetin, buddlejasaponin IV, acetylSSc, 4′′-*O*-AcetylSSa, SSe, 6′′-*O*-AcetylSSa, and 6′′-*O*-AcetylSSd) appeared. Meanwhile SSg, 3′′-*O*-AcetylSSa, and SSd showed the same tendency and this would result in the unidentifiable problem.

For PRA, 20 peaks showed significant differences after processing. Comparing with PRA, the intensity of 13 peaks enhanced in VPPRA, the other seven peaks decreased in VPPRA. Considering compatibility, 10 of these 20 peaks disappeared in the herb-pair, at the same time, 6-*O*-β-d-glucopyranosyl lactinolide and benzoylpaeoniflorin appeared. Also, cianidanol enhanced in individual herb but decreased in herb-pair. These changes perhaps resulted from compatibility. Moreover, nine peaks had the same trend after processing of PRA in individual herb and herb-pair, and it was also hard to distinguish as BR. Under further influence of formula compatibility with AFI and GRM, five peaks showing significant differences in herb-pair vanished in formula; oppositely, five new peaks (oxypaeoniflora, 6′-*O*-β-d-glucopyranosylalbiflorin, galloylpaeoniflorin, lactiflorin, benzoylalbiflorin) appeared. Formula compatibility may be responsible for these changes. In addition, seven peaks (6-*O*-β-d-glucopyranosyl lactinolide, mudanpioside f, albiflorin, isomaltoalbiflorin, paeoniflorigenone, paeonol, and benzoic acid) displayed an identical trend; this still led to the unidentifiable problem. Figure 3 shows the comparison of the contents of the components identified with significant differences. Processing with vinegar and formula compatibility can both regulate the acidity and alkalinity of the solution and promote changes in chemical composition, such as hydrolysis reaction, isomerization reaction, etc., resulting in increased or decreased dissolution of some components. Finally, we found that processing of BR and PRA also had the impact on AFI and GRM, and the results were shown in Table 4.

As shown in Table 2, the intensity of paeonol significantly increased after stir-frying with vinegar. According to a previous report [16], adding acid could greatly improve the extraction efficiency of paeonol. Since the boiling point of paeonol is 154 °C, the use of slow fire (130 °C) controlled by infrared radiation thermometer during the processing minimized the loss of paeonol. In addition, acetic acid plays an important role to form intermolecular hydrogen bonds by Van der Waals’ force with paeonol, resulting in the increase of dissolution rate. Modern researches indicate that paeonol has analgesic and antiphlogistic pharmacological activities [17,18] and is consistent with TCM theory that processing of medicinal herbs with vinegar can enhance the effects of promoting blood circulation and relieving pain. As an illustration, Figure 4 revealed the course of deducing fragmentation of paeonol.

As shown in Table 3, we found that the intensity of SSa and SSd declined but the intensity of SSb_2_ and SSb_1_ increased in the BR. SSs, a kind of oleanane type triterpenoid saponin, could be divided into seven types according to their different aglycones. SSa, SSd, and SSc are epoxy-ether saikosaponins (type I), while SSb_2_ and SSb_1_ with a different aglycone, form a heterocyclic diene saikosaponin (type II) [19]. The glycosidic bond is very easily hydrolyzed in the acidic conditions or being heated [20,21]. Vinegar processing could promote the hydrolyzation from 13 to 28 allyl oxide linkage to its corresponding heteroannular diene structure, resulting in the aglycone accumulation. As shown in Figure 5, peak No. 94 was clearly observed in VPBR, VPBR-VPPRA herb-pair, SNS-containing VPBR and VPPRA, and SNS, and almost undetectable in BR and BR-PRA herb-pair. According to the fragmentations in both positive and negative ion modes and other reports [22,23,24], we suggested that peak No. 94 is SSg. SSg in SNS could be related to the acidic compounds of herbal formula, such as glycyrrhizic acid. Also, peak No. 68 (SSh/i), as the isomer of SSc, had the same change with SSg. Based on these, we hypothesized that SSa and SSd could be transformed to SSb_2_, SSb_1_, and SSg, while SSc could be converted to SSh and SSi after processing and formula compatibility.

## 3. Materials and Methods

### 3.1. Materials and Reagents

Acetonitrile (Merck, Darmstadt, Germany) and formic acid from Anaqua Chemical Supply (ACS, Houston, TX, USA) of HPLC/MS-grade were purchased for UHPLC-Q-TOF/MS analysis. Deionized water was prepared using a Milli-Q system (Millipore, Molsheim, France). SPE columns (LC-C_18_, 500 mg/mL) were purchased from ANPLE Scientific Instrument (Shanghai, China). Other reagents of analytical grade were purchased from Nanjing Chemical Reagent Co., Ltd. (Jiangsu, China).

BR, PRA, AFI, and GRM were obtained from different Chinese pharmacies and pharmaceutical factories, and authenticated by Professor Hao Cai. The quality of all collected samples was strictly evaluated and consistent with the regulations of Chinese Pharmacopoeia (Edition 2015, Part One). VPBR and VPPRA were prepared according to the processing standards described in Chinese Pharmacopoeia (Edition 2015, Part Four). The voucher specimens were deposited in School of Pharmacy, Nanjing University of Chinese Medicine (Nanjing, China).

### 3.2. Sample Preparation

The decoction of BR was prepared as follows. Eight grams of BR were extracted twice in a reflux water heating mantle in 48 mL and 32 mL of deionized water for 1.5 h and 1 h of reflux, respectively. The mixed solution was filtered through a four-layer mesh following the reflux. One milliliter of the solution was loaded onto a C_18_ RP SPE column and the gradient elution was performed as the following sequence. One milliliter of 20% acetonitrile in water (20:80, *v*/*v*), 1 mL of 40% acetonitrile in water (40:60, *v*/*v*), 1 mL of 60% acetonitrile in water (60:40, *v*/*v*), 1 mL of 80% acetonitrile in water (80:20, *v*/*v*), and 1 mL of acetonitrile. After the sequent elution, the collected eluent was eddied for 2 min and centrifuged at 13,000 rpm for 5 min. Finally, the supernatant was collected as the injection solution. The decoctions of VPBR, PRA, and VPPRA were prepared according to the same procedures above.

The decoction of BR-PRA herb-pair consisted of 4 g of BR and 4 g of PRA, and prepared as the same procedures as individual herb described above. The decoction of VPBR-VPPRA herb-pair was prepared using the same procedures as the decoction of BR-PRA herb-pair. The decoction of SNS was consist of 2 g of BR, 2 g of PRA, 2 g of AFI, and 2 g of GRM, and prepared using the same procedures as individual herb. The decoction of SNS containing VPBR and VPPRA was prepared using the same procedures as the decoction of SNS.

### 3.3. Chromatographic Separation

Chromatographic analysis was performed using a UHPLC system (Shimadzu, Kyoto, Japan) consisting of an LC-30AD binary pump, an autosampler (Model SIL-30SD), an online degasser (DGU-20A5R), and a temperature controller for columns (CTO-30A). Separation was carried out on an extended C_18_ Column (2.1 mm × 100 mm, 1.8 μm; Agilent, Palo Alto, CA, USA) at 30 °C and the flow rate was 0.3 mL/min. The optimal mobile phase consisted of A (HCOOH/H_2_O, 0.1:100, *v*/*v*) and B (C_2_H_3_N). The optimized UHPLC elution conditions were as follows 0–2 min, 3–15% B; 2–7 min, 15–20% B;7–8 min, 20% B; 8–9 min, 20–30% B; 9–13 min, 30–32% B; 13–21 min; 32–54% B; 21–23 min, 54–100% B; 23–27 min, 100–3% B; and 27–28 min, 3% B. The injection volume was 2 μL.

### 3.4. MS and MS/MS Experiments

A triple TOF 5600^+^ System (AB Sciex, Concord, CA, USA) equipped with an electrospray ionization (ESI) source was performed. The MS was operated in both positive and negative ion modes. Parameters were set as follows: ion spray voltage of +4500/−4500 V; turbo spray temperature of 550 °C; declustering potential (DP) of +60/−60 V; collision energy of +35/−45 V; nebulizer gas (gas 1) of 55 psi; heater gas (gas 2) of 55 psi and curtain gas of 35 psi. TOF MS and TOF MS/MS were scanned with the mass ranges of *m*/*z* 100–2000 and 50–1000, respectively. The experiments were run with 200 ms accumulation time for TOF MS and 80 ms accumulation time for TOF MS/MS. Continuous recalibration was performed at the intervals of 3 h. Dynamic background subtraction and information-dependent acquisition techniques were applied to reduce the impact of matrix interference and increase the efficiency of analysis.

### 3.5. MS and MS/MS Data Processing and Analysis

The raw data were obtained by the Analyst TF 1.6 software (AB Sciex, Concord, CA, USA). Before data processing, a database about chemical components of medicinal herbs in SNS, including names, molecular formulas, chemical structures, and accurate molecular weights, was established by searching relevant reported literature and database websites, including PubMed and SciFinder. The data were analyzed by using PeakView^TM^ 1.2 software (AB Sciex, Concord, CA, USA) for a perfect match with the information in the established database, according to fragmentations of the different peaks. The main parameters used were set as follows: retention time range of 0–28 min, mass range of 100 to 2000 Da, and mass tolerance of 10 ppm. By using the method of PCA with MarkerView^TM^ 1.2.1 software (AB Sciex, Concord, CA, USA) to check for outliers and variation trend, the gathered data were more intuitionistic. The Student’s *t*-test was performed to find out a list of peaks that were finally defined as the main contributors to the significant difference between raw and processed medicinal herbs (*p* < 0.05).

## 4. Conclusions

A total of 122 constituents had been identified by creative global analysis in individual herb, herb-pair, and complicated Chinese herbal formula of SNS. Taking BR as an example, 29 kinds of SSs had been identified, including some new discoveries in recent years, such as SSq, SSm, and so forth. Monoterpene glycosides (oxypaeoniflora, mudanpioside f, paeoniflorigenone, etc) showed a marked increase after processing of PRA. This is the first report of SSh/i and SSg being identified in SNS. Through three progressive levels of comparison, it suggests that processing herbal medicine and/or changing medicinal formula compatibility could alter herbal chemical constituents, resulting in different pharmaceutical effects. Herbal formula has always been the predicament of Chinese medicine research, and some scholars only employed SSd and paeoniflorin (the main components of BR and PRA) for research [25], whereas the effects between individual components and herbal formula containing individual components are quite different. We hope that the thoughts of this article would be some helpful for further research of herbal formula.

## Figures and Tables

**Figure 1 molecules-23-03128-f001:**
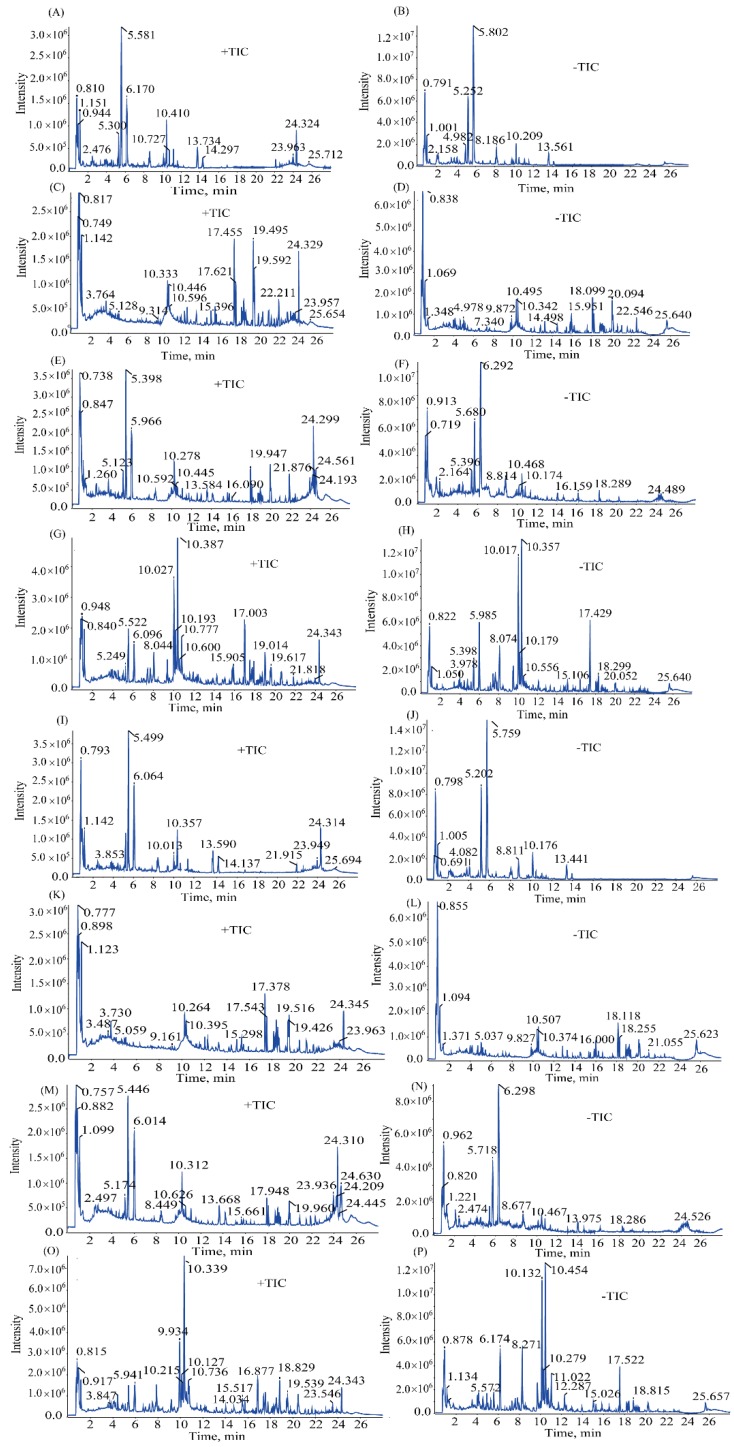
Typical total ion chromatograms (TICs) in positive ion mode of PRA (**A**), VPPRA (**I**), BR (**C**), VPBR (**K**), BR-PRA herb-pair (**E**), VPBR-VPPRA herb-pair (**M**), SNS (**G**), and SNS-containing VPBR and VPPRA (**O**). Typical total ion chromatograms (TICs) in negative ion mode of PRA (**B**), VPPRA (**J**), BR (**D**), VPBR (**L**), BR-PRA herb-pair (**F**), VPBR-VPPRA herb-pair (**N**), SNS (**H**), and SNS-containing VPBR and VPPRA (**P**).

**Figure 2 molecules-23-03128-f002:**
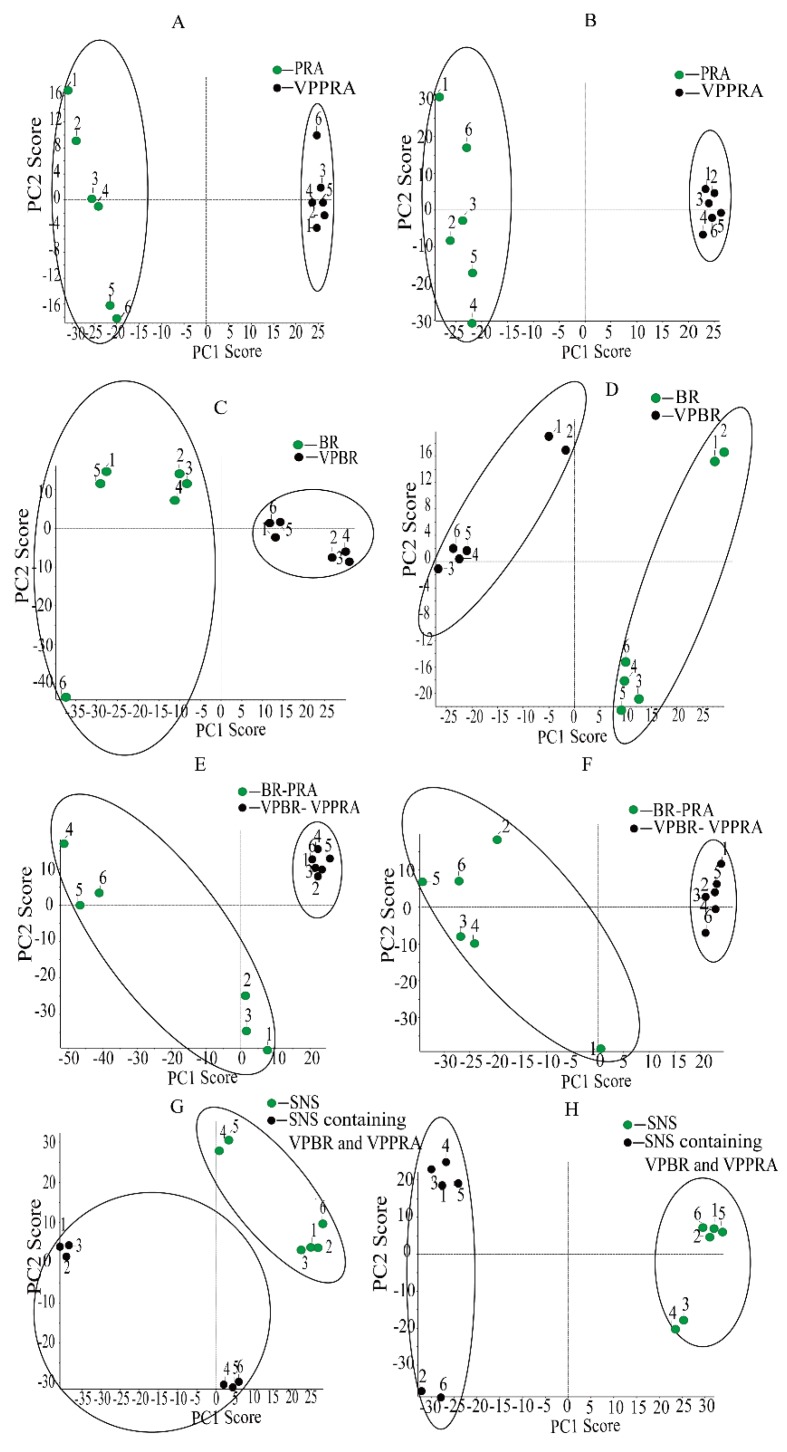
Principal component analysis (PCA) score plots in positive ion mode of PRA and VPPRA (**A**), BR and VPBR (**C**), BR-PRA herb-pair and VPBR-VPPRA herb-pair (**E**), SNS and SNS containing VPBR and VPPRA (**G**). PCA score plots in negative ion mode of PRA and VPPRA (**B**), BR and VPBR (**D**), BR-PRA herb-pair and VPBR-VPPRA herb-pair (**F**), and SNS and SNS-containing VPBR and VPPRA (**H**).

**Figure 3 molecules-23-03128-f003:**
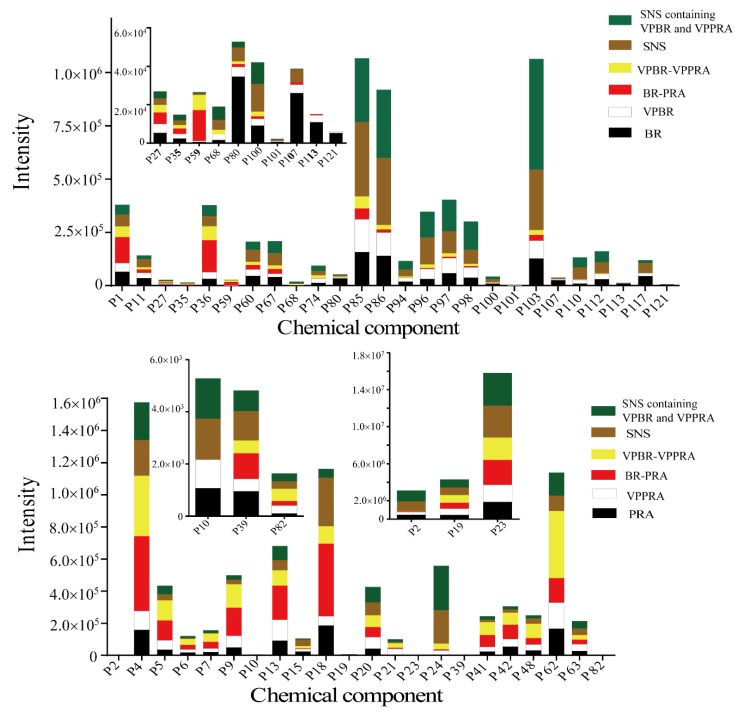
Contents of components identified with significant differences in individual herb, herb-pair, and complicated Chinese herbal formula of SNS.

**Figure 4 molecules-23-03128-f004:**
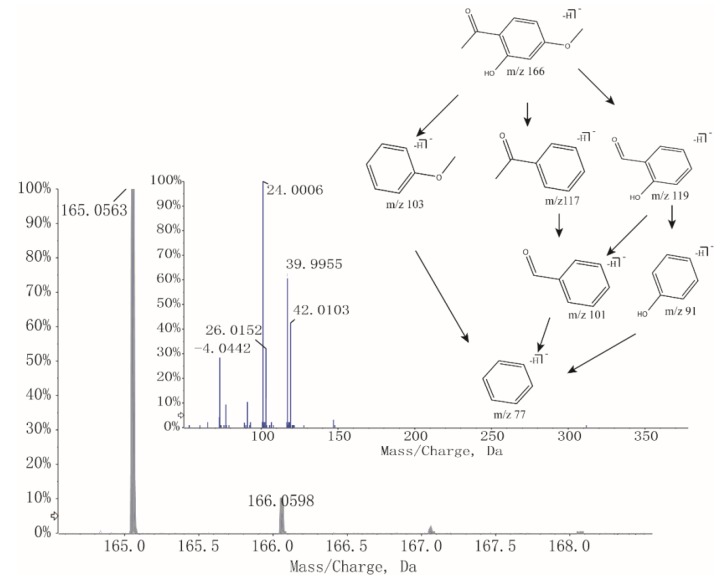
MS and tandem mass spectroscopy (MS/MS) spectra and fragmentation of Paeonol.

**Figure 5 molecules-23-03128-f005:**
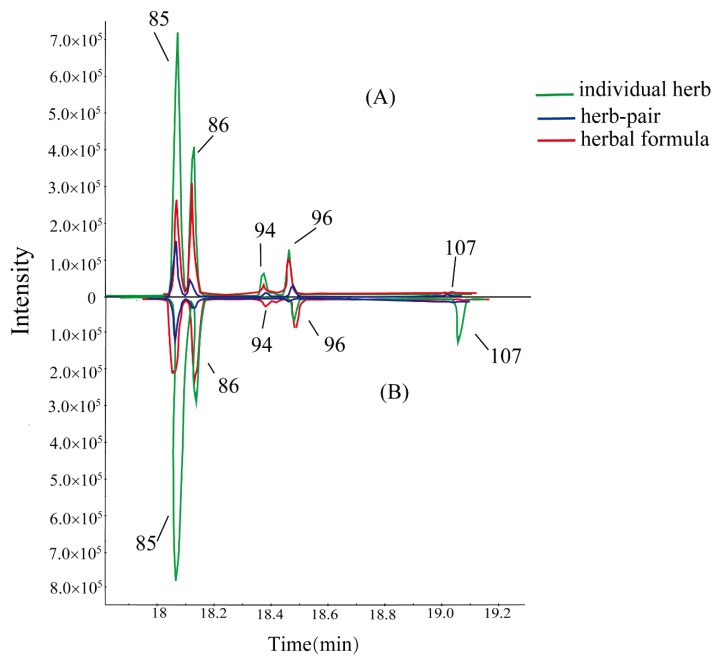
Comparison on intensity of five isomers of Saikosaponins in BR, BR-PRA herb-pair, and SNS (**B**). Comparison on intensity of five isomers of Saikosaponins in VPBR, VPBR-VPPRA herb-pair, and SNS-containing VPBR and VPPRA (**A**).

**Table 1 molecules-23-03128-t001:** Identification of chemical compounds by ultrahigh performance liquid chromatography coupled with electrospray ionization tandem quadrupole-time-of-flight mass spectrometry (UHPLC-Q-TOF-MS/MS).

No.	Compound	T_R_ (min)	Molecular Formula	Detected Mass (*m*/*z*) Ion Type	Mass Error (ppm)	MS/MS *(m*/*z*)	Purity Score	Source
1	Adonitol	0.82	C_5_H_12_O_5_	151.0612 [M − H]^−^	4.0	101.0283,83.0141,71.0171	88.30%	BR
2	Sucrose	0.83	C_12_H_22_O_11_	387.1133 [M + HCOOH-H]^−^	1.0	341.1099,161.0458,89.0264	100.00%	PRA
3	Synephrine	1.22	C_9_H_13_NO_2_	168.1019 [M + H]^+^	−2.4	150.0912,121.0653,91.0553	76.20%	AFI
4	Gallic acid	1.97	C_7_H_6_O_5_	169.0142 [M − H]^−^	2.4	125.0246,79.0210,51.0283	79.40%	PRA
5	1-*O*-β-d-glucopyranosyl-paeonisuffrone	2.39	C_16_H_24_O_9_	405.1391 [M + HCOOH-H]^−^	1.3	197.0814,137.0603,85.0304	83.90%	PRA
6	6-*O*-β-d-glucopyranosyl lactinolide	3.16	C_16_H_26_O_9_	407.1547 [M + HCOOH-H]^−^	1.4	361.1514,199.0974,101.0250	100.00%	PRA
7	Mudanpioside f	3.31	C_16_H_24_O_8_	389.1442 [M + HCOOH-H]^−^	0.9	181.0835,151.0767,109.0646	88.10%	PRA
8	Neochlorogenic acid	3.48	C_16_H_18_O_9_	353.0884 [M − H]^−^	1.6	191.0555, 135.0449, 85.0305	93.90%	BR
9	Oxypaeoniflora	3.83	C_23_H_28_O_12_	495.1508 [M − H]^−^	−0.9	495.1560,137.0238	100.00%	PRA
10	4′′-Hydroxy-3′′-methoxyalbiflorin	4.16	C_24_H_30_O_13_	571.1658 [M + HCOOH-H]^−^	−2.0	525.1653,363.1080,167.0345	100.00%	PRA
11	Chlorogenic acid	4.18	C_16_H_18_O_9_	353.0878 [M − H]^−^	0.9	191.0558,85.0300	100.00%	BR
12	Cryptochlorogenic acid	4.35	C_16_H_18_O_9_	353.0888 [M − H]^−^	2.7	191.0554, 155.0330, 93.0353	93.50%	BR
13	Cianidanol	4.40	C_15_H_14_O_6_	289.0718 [M − H]^−^	2.0	245.0824,137.0238,109.0294	81.10%	PRA
14	Fabiatrin	4.52	C_21_H_26_O_13_	531.1345 [M + HCOOH-H]^−^	2.3	177.0189	100.00%	AFI
15	6′-*O*-β-d-glucopyranosylalbiflorin	4.76	C_29_H_38_O_16_	687.2131 [M + HCOOH-H]^−^	−8.7	641.2078,489.1612,183.0668	100.00%	PRA
16	5,7-dihydroxycoumarin	4.99	C_9_H_6_O_4_	177.0193 [M − H]^−^	6.3	177.0205,69.0003	72.80%	AFI
17	Lonicerin	5.03	C_27_H_30_O_15_	593.1512 [M − H]^−^	−0.7	593.1543,353.0670,297.0776	100.00%	AFI
18	Isomaltopaeoniflorin	5.13	C_29_H_38_O_16_	687.2131 [M + HCOOH-H]^−^	0.2	611.2036,323.0995,165.0563	100.00%	PRA
19	Albiflorin	5.49	C_23_H_28_O_11_	481.1704 [M + H]^+^	1.0	319.1183,133.0645,105.0334	100.00%	PRA
20	Paeoniflorigenone	5.58	C_17_H_18_O_6_	319.1176 [M + H]^+^	2.5	151.0757,105.0349,77.0406	100.00%	PRA
21	Isomaltoalbiflorin	5.61	C_29_H_38_O_16_	687.2131 [M + HCOOH-H]^−^	−9.1	641.2088,491.1763	100.00%	PRA
22	Schaftoside	5.91	C_26_H_28_O_14_	563.1406 [M − H]^−^	1.0	563.1478,443.1016,365.0682	100.00%	GRM
23	Paeoniflorin	5.98	C_23_H_28_O_11_	525.1603 [M + HCOOH-H]^−^	5.3	449.1492,165.0558,121.0301	100.00%	PRA
24	Paeonol	6.70	C_9_H_10_O_3_	165.0557 [M − H]^−^	1.8	119.0507,96.9579	100.00%	PRA
25	Ethyl gallate	6.90	C_9_H_10_O_5_	197.0456 [M − H]^−^	2.5	162.8362,89.0271,59.0154	100.00%	PRA
26	SSq	7.32	C_54_H_88_O_24_	1165.5639 [M + HCOOH-H]^−^	0.2	1119.5784, 1089.5630	91.00%	BR
27	Rutin	7.57	C_27_H_30_O_16_	611.1607 [M + H]^+^	−1.4	303.0506	100.00%	BR
28	liquiritin apioside	7.75	C_26_H_30_O_13_	549.1614 [M − H]^−^	1.2	549.1659,255.0666,135.0088	100.00%	GRM
29	Liquiritin	7.93	C_21_H_22_O_9_	417.1191 [M − H]^−^	0.5	255.0667,135.0090	100.00%	GRM
30	Neoeriocitrin	8.18	C_27_H_32_O_15_	595.1668 [M − H]^−^	0.8	595.1719,287.0566,135.0449	100.00%	AFI
31	Scopoletin	9.04	C_10_H_8_O_4_	237.0394 [M + HCOOH-H]^−^	6.7	121.0295,93.0328,71.0160	100.00%	AFI
32	Kaempferol	9.19	C_15_H_10_O_6_	287.0550 [M + H]^+^	−0.7	287.0533,93.0374	100.00%	BR
33	SSv	9.47	C_53_H_86_O_24_	1151.5480 [M + HCOOH-H]^−^	−3.1	1105.5579,791.4285,313.1119	95.90%	BR
34	Narirutin	9.50	C_27_H_32_O_14_	579.1719 [M − H]^−^	2.9	271.0622,151.0033	100.00%	AFI
35	Isorhamnetin	9.56	C_16_H_12_O_7_	317.0656 [M + H]^+^	0.2	317.0648,257.0430	73.60%	BR
36	Isorhamnetin-3-rutinoside	9.65	C_28_H_32_O_16_	623.1618 [M − H]^−^	−0.1	623.1665,315.0513,299.0196	75.50%	BR
37	Isochlorogenic acid b	9.68	C_25_H_24_O_12_	561.1239 [M + HCOOH-H]^−^	0.3	385.0916,193.0504,147.0257	83.60%	BR
38	Naringin	10.07	C_27_H_32_O_14_	579.1719 [M − H]^−^	0.9	579.1771,271.0614,151.0032	100.00%	AFI
39	Benzoic acid	10.08	C_7_H_6_O_2_	123.0441 [M + H]^+^	−5.6	105.0358,77.0394	90.50%	PRA
40	Isorhamnetin-3-glucoside	10.11	C_22_H_22_O_12_	479.1184 [M + H]^+^	−2.0	317.0667	89.10%	BR
41	Mudanpioside i	10.18	C_23_H_28_O_11_	479.1559 [M − H]^−^	−1.5	121.0302,77.0416	100.00%	PRA
42	Galloylpaeoniflorin	10.19	C_23_H_28_O_10_	509.1654 [M + HCOOH-H]^−^	1.1	121.0302,77.0415	100.00%	PRA
43	Neohesperidin	10.23	C_28_H_34_O_15_	609.1825 [M − H]^−^	1.4	325.0730,301.0726	70.40%	AFI
44	Hesperetin	10.39	C_16_H_14_O_6_	303.0863 [M + H]^+^	1.4	303.0872,153.0181,67.0204	100.00%	AFI
45	Hesperidin	10.40	C_28_H_34_O_15_	609.1825 [M − H]^−^	0.8	609.1887,301.0725,283.0621	72.60%	AFI
46	Isoliquiritin apioside	10.56	C_26_H_30_O_13_	549.1614 [M − H]^−^	2.3	549.1667,255.0663,135.0082	100.00%	GRM
47	Isochlorogenic acid a	10.61	C_25_H_24_O_12_	561.1239 [M + HCOOH-H]^−^	−0.9	323.0849,193.0482,147.0296	82.60%	BR
48	Lactiflorin	10.68	C_23_H_26_O_10_	480.1864 [M + NH4]^+^	0.5	301.1076,151.0752,105.0343	100.00%	PRA
49	Ononin	10.83	C_22_H_22_O_9_	431.1337 [M + H]^+^	1.1	269.0807	100.00%	GRM
50	Rhoifolin	10.94	C_27_H_30_O_14_	577.1563 [M − H]^−^	−0.9	271.0613,151.0030	100.00%	AFI
51	Isochlorogenic acid c	11.06	C_25_H_24_O_12_	561.1239 [M + HCOOH-H]^−^	0.2	323.0766,193.0494,147.0452	75.70%	BR
52	Clinoposaponin XII	11.19	C_42_H_68_O_14_	795.4536 [M − H]^−^	−0.4	795.4661,633.4072,471.3084	100.00%	BR
53	epinortrachelogenin	11.86	C_20_H_22_O_7_	373.1293 [M − H]^−^	0.3	179.0711,99.0091	70.90%	BR
54	Heraclenin	11.93	C_16_H_14_O_5_	287.0914 [M + H]^+^	0.9	287.0906,153.0176,133.0640	100.00%	AFI
55	Liquiritigenin	11.98	C_15_H_12_O_4_	255.0663 [M − H]^−^	2.6	135.0082,119.0505,91.0195	100.00%	GRM
56	HOSSa	12.17	C_42_H_70_O_14_	797.4693 [M − H]^−^	−1.9	635.4196	100.00%	BR
57	Puerarin	12.29	C_21_H_20_O_9_	417.1180 [M + H]^+^	−0.2	417.1094,367.0811,131.0498	77.70%	BR
58	5,4′′-dihydroxy-3,7-dimethoxyflavone	12.65	C_17_H_14_O_6_	315.0863 [M + H]^+^	0.2	315.0856,243.0647,175.0386	72.50%	GRM
59	HOSSd	12.79	C_42_H_70_O_14_	797.4693 [M − H]^−^	−0.9	635.4235	100.00%	BR
60	Buddlejasaponin IV	13.24	C_48_H_78_O_18_	987.5159 [M + HCOOH-H]^−^	0.3	941.5229,795.4616	100.00%	BR
61	Clinoposaponin XIV	13.45	C_42_H_68_O_14_	795.4536 [M − H]^−^	0.0	795.4627,633.3986,457.3314	100.00%	BR
62	Benzoylpaeoniflorin	13.96	C_30_H_32_O_12_	629.1864 [M + HCOOH-H]^−^	1.0	165.0562,121.0307	100.00%	PRA
63	Benzoylalbiflorin	14.09	C_30_H_32_O_12_	585.1967 [M + H]^+^	−1.0	319.1172,197.0798,133.0643	100.00%	PRA
64	Licoricesaponin A_3_	14.83	C_48_H_72_O_21_	983.4493 [M − H]^−^	0.2	983.4633,497.1162	100.00%	GRM
65	(+/−)−Naringenin	14.85	C_15_H_12_O_5_	271.0612 [M − H]^−^	2.7	187.0396,119.0511	100.00%	AFI
66	4,4′-dihydroxy-2-methoxychalcone	15.30	C_16_H_14_O_4_	269.0819 [M − H]^−^	4.4	269.0707,133.0297,117.0337	73.50%	GRM
67	SSc	15.79	C_48_H_78_O_17_	971.5210 [M + HCOOH-H]^−^	−0.7	925.5193,779.4675	100.00%	BR
68	SSi/h	15.90	C_48_H_78_O_17_	971.5209 [M + HCOOH-H]^−^	−0.1	925.5296, 779.4640	100.00%	BR
69	Salicifoline	16.13	C_20_H_20_O_6_	355.1187 [M − H]^−^	−2.7	184.9549,129.0726	85.90%	BR
70	Licoricesaponin G_2_	16.42	C_42_H_62_O_17_	837.3914 [M − H]^−^	−0.5	837.4008,351.0573,193.0347	96.60%	GRM
71	Deacetylnomilinic acid	16.50	C_26_H_34_O_9_	489.2130 [M − H]^−^	−1.6	489.2174,333.1706,203.0687	93.20%	AFI
72	Licoricesaponin E_2_	16.55	C_42_H_60_O_16_	819.3809 [M − H]^−^	−0.9	819.3925,351.0577,193.0343	100.00%	GRM
73	Enoxolone	16.61	C_30_H_46_O_4_	471.3469 [M + H]^+^	0.5	471.3489,219.1769,177.1636	81.70%	GRM
74	SSh/i	16.62	C_48_H_78_O_17_	971.5210 [M + HCOOH-H]^−^	0.9	925.5193	100.00%	BR
75	Licoricesaponin D_3_	16.64	C_50_H_76_O_21_	1011.4806 [M − H]^−^	−0.9	1011.4976,497.1175	100.00%	GRM
76	SSb_3_/b_4_	16.96	C_43_H_72_O_14_	857.4893 [M + HCOOH-H]^−^	−4.1	811.4911,649.4320,161.0409	92.40%	BR
77	Glycyrrhizic acid	17.00	C_42_H_62_O_16_	823.4111 [M + H]^+^	0.9	647.3782,471.3467,453.3356	100.00%	GRM
78	Isoliquiritigenin	17.09	C_15_H_12_O_4_	255.0663 [M − H]^−^	0.4	135.0074,119.0495,91.0186	100.00%	GRM
79	Formononetin	17.31	C_16_H_12_O_4_	269.0808 [M + H]^+^	1.0	269.0811,197.0600	81.80%	GRM
80	Acetyl-SSc	17.32	C_50_H_80_O_18_	1013.5316 [M + HCOOH-H]^−^	−2.8	967.5370,779.4628	96.60%	BR
81	Betulonicacid	17.68	C_30_H_46_O_3_	455.3520 [M + H]^+^	0.0	455.3525,285.2216,133.1008	82.80%	GRM
82	Palbinone	17.78	C_22_H_30_O_4_	357.2071 [M − H]^−^	−1.7	357.2067,285.1906,241.1612	92.40%	PRA
83	SSn	17.83	C_48_H_78_O_18_	987.5156 [M + HCOOH-H]^−^	−0.3	941.5220, 779.4644	100.00%	BR
84	SSm/e	17.94	C_42_H_68_O_12_	809.4682 [M + HCOOH-H]^−^	−4.8	763.4729, 617.4095, 161.0454	86.50%	BR
85	SSa	18.11	C_42_H_68_O_13_	825.4631 [M + HCOOH-H]^−^	2.8	779.4587, 617.4059	100.00%	BR
86	SSb_2_	18.25	C_42_H_68_O_13_	825.4631 [M + HCOOH-H]^−^	2.5	779.4587, 617.4059	100.00%	BR
87	Licoricesaponin K_2_	18.35	C_42_H_62_O_16_	821.3965 [M − H]^−^	0.3	821.4084,351.0578,193.0350	72.00%	GRM
88	Licoricesaponin H_2_	18.64	C_42_H_62_O_16_	821.3965 [M − H]^−^	−0.4	821.4067,351.0582	100.00%	GRM
89	Limonin	18.75	C_26_H_30_O_8_	469.1868 [M − H]^−^	−2.7	469.1872,229.1219,145.0650	90.20%	AFI
90	2′′-*O*-Acetyl-SSa	18.82	C_44_H_70_O_14_	867. 4737 [M + HCOOH-H]^−^	−0.3	821.4798,779.4684,617.4118	94.90%	BR
91	Nomilinic acid	18.85	C_28_H_36_O_10_	531.2236 [M − H]^−^	−1.2	489.2170,325.1799,59.0169	100.00%	AFI
92	Dipropyl phthalate	18.86	C_14_H_18_O_4_	249.1132 [M − H]^−^	3.0	149.0935,59.0177	85.70%	PRA
93	Licoricesaponin J_2_	18.89	C_42_H_64_O_16_	823.4122 [M − H]^−^	0.1	823.4212,351.0575,193.0352	100.00%	GRM
94	SSg	18.98	C_42_H_68_O_13_	825.4631 [M + HCOOH-H]^−^	1.3	779.4665,617.4099	100.00%	BR
95	Nobiletin	19.04	C_21_H_22_O_8_	403.1387 [M + H]^+^	1.1	403.1383,373.0912,327.0860	77.80%	AFI
96	SSb_1_	19.05	C_42_H_68_O_13_	825.4631 [M + HCOOH-H]^−^	1.1	779.4587,617.4059	100.00%	BR
97	3′′-*O*-Acetyl-SSa	19.13	C_44_H_70_O_14_	867. 4737 [M + HCOOH-H]^−^	0.2	821.4781,779.4662,617.4096	100.00%	BR
98	4′′-*O*-Acetyl-SSa	19.28	C_44_H_70_O_14_	867. 4737 [M + HCOOH-H]^−^	−0.7	821.4775,779.4658,617.4094	100.00%	BR
99	Licoricesaponin C_2_	19.44	C_42_H_62_O_15_	805.4016 [M − H]^−^	0.1	805.4118,351.0568	100.00%	GRM
100	prosaikogenin f	19.49	C_36_H_58_O_8_	663.4103 [M + HCOOH-H]^−^	−0.6	617.4094,145.0499	77.80%	BR
101	SSe/m	19.59	C_42_H_68_O_12_	809.4682 [M + HCOOH-H]^−^	−1.3	763.4722,601.4170,161.0442	96.40%	BR
102	Licoricesaponin B_2_	19.72	C_42_H_64_O_15_	807.4173 [M − H]^−^	−0.6	807.4277,351.0574,193.0343	100.00%	GRM
103	6′′-*O*-Acetyl-SSa	20.09	C_44_H_70_O_14_	867. 4737 [M + HCOOH-H]^−^	−0.2	821.4780,779.4666,617.4095	100.00%	BR
104	Licoisoflavone a	20.27	C_20_H_18_O_6_	353.1031 [M − H]^−^	2.4	353.1056,285.1131,171.0446	74.90%	GRM
105	Glycycoumarin	20.40	C_21_H_20_O_6_	367.1187 [M − H]^−^	0.5	367.1188,309.0411,201.0187	89.90%	GRM
106	Prosaikogenin g	20.56	C_36_H_58_O_8_	663.4103 [M + HCOOH-H]^−^	−0.1	617.4060,145.0540	100.00%	BR
107	SSd	20.61	C_42_H_68_O_13_	825.4631 [M + HCOOH-H]^−^	2.5	779.4587,617.4059	100.00%	BR
108	Sinensitin	20.66	C_20_H_20_O_7_	373.1282 [M + H]^+^	1.1	373.1288,297.0766	79.80%	AFI
109	Diacetyl-SSd	21.03	C_46_H_72_O_15_	909.4823 [M + HCOOH-H]^−^	−0.7	863.4894,821.4782,761.4554	92.00%	BR
110	2′′-*O*-Acetyl-SSd	21.04	C_44_H_70_O_14_	867. 4737 [M + HCOOH-H]^−^	−0.4	821.4766,779.4660,617.4085	100.00%	BR
111	Liconeolignan	21.33	C_21_H_22_O_5_	354.1467 [M − H]^−^	−1.9	353.1020,297.0441,173.0224	80.30%	GRM
112	Diacetyl-SSd	21.61	C_46_H_72_O_15_	909.4823 [M + HCOOH-H]^−^	0.0	863.4896,821.4773,761.4552	100.00%	BR
113	3′′-*O*-Acetyl-SSd	21.89	C_44_H_70_O_14_	867. 4737 [M + HCOOH-H]^−^	0.3	821.4794,779.4683,617.4103	100.00%	BR
114	Acetyl-SSe	21.95	C_44_H_70_O_13_	851.4788 [M + HCOOH-H]^−^	−1.7	805.4838,763.4701,601.4155	91.00%	BR
115	Neoglycyrol	22.36	C_21_H_18_O_6_	365.1031 [M − H]^−^	0.3	365.1037,307.0250,207.0430	96.40%	GRM
116	Prosaikogenin d	22.42	C_36_H_58_O_8_	663.4103 [M + HCOOH-H]^−^	−1.9	617.408	72.80%	BR
117	6′′-*O*-Acetyl-SSd	22.54	C_44_H_70_O_14_	867. 4737 [M + HCOOH-H]^−^	−0.9	821.4770,779.4651,617.4087	100.00%	BR
118	Obacunon	22.74	C_26_H_30_O_7_	453.1919 [M − H]^−^	−4.2	453.2044,339.1957,149.0963	90.20%	AFI
119	Saikogenin e	22.78	C_30_H_48_O_3_	455.3519 [M − H]^−^	−2.5	455.3529, 325.1855, 152.9936	93.20%	BR
120	Diacetyl-SSd	23.06	C_46_H_72_O_15_	909.4823 [M + HCOOH-H]^−^	−3.0	863.4894,821.4774,617.4091	100.00%	BR
121	Diacetyl-SSd	23.24	C_46_H_72_O_15_	909.4823 [M + HCOOH-H]^−^	−2.7	863.4859,821.4747,761.4478	94.40%	BR
122	Saikogenin f	23.38	C_30_H_48_O_4_	533.3473 [M + HCOOH-H]^−^	−6.3	471.3452,453.1727,388.9749	90.10%	BR

**Table 2 molecules-23-03128-t002:** Results of the *t*-test of 26 peaks from BR showing significant difference in individual herb, herb-pair, and complicated Chinese herbal formula before and after processing (*n* = 6).

BR			Individual Herb	Herb-Pair	Herbal Formula
No.	T_R_ (min)	Identified Compound	*p*-Value	*p*-Value	*p*-Value
1	0.82	Adonitol	0.00394 ↓ **	0.00011 ↓ **	
11	4.18	Chlorogenic acid	0.00137 ↓ **		
27	7.57	Rutin	0.00946 ↓ **		
35	9.56	Isorhamnetin	0.04005 ↑ *		0.01029 ↑ *
36	9.65	Isorhamnetin-3-rutinoside		0.00055 ↓ **	
59	12.79	HOSSd		0.00077 ↓ **	
60	13.24	Buddlejasaponin IV	1.72 × 10^−5^ ↓ **		3.90 × 10^−6^ ↓ **
67	15.79	SSc	0.01180 ↓ *		
68	15.9	SSi/h	0.00130 ↑ **	3.34 × 10^−8^ ↑ **	
74	16.62	SSh/i	0.00017 ↑ **	0.00027 ↑ **	
80	17.32	AcetylSSc	0.00089 ↓ **		0.01867 ↓ *
85	18.11	SSa	0.00475 ↓ **		
86	18.25	SSb_2_	0.03997 ↑ *		
94	18.98	SSg	0.00577 ↑ **	1.90 × 10^−6^ ↑ **	0.04480 ↑ *
96	19.05	SSb_1_	0.00656 ↑ **	4.85 × 10^−5^ ↑ **	
97	19.13	3′′-*O*-AcetylSSa	0.00016 ↑ **	4.18 × 10^−6^ ↑ **	0.002821 ↑ **
98	19.28	4′′-*O*-AcetylSSa			0.001645 ↑ **
100	19.49	prosaikogenin f	0.00031 ↓ **	0.00281 ↑ **	
101	19.59	SSe/m	0.00014 ↓ **		0.00626 ↓ **
103	20.09	6′′-*O*-AcetylSSa	4.43 × 10^5^ ↓ **		0.024542 ↓ *
107	20.61	SSd	0.00299 ↓ **	0.00078 ↓ **	0.04567 ↓ *
110	21.04	2′′-*O*-AcetylSSd		0.00116 ↓ **	
112	21.61	Diacetyl-SSd	0.03744 ↓ *		
113	21.89	3′′-*O*-AcetylSSd	9.31 × 10^−7^ ↓ **		
117	22.54	6′′-*O*-AcetylSSd	0.00053 ↓ **		0.04310 ↓ *
121	23.24	Diacetyl-SSd	1.06 × 10^−8^ ↓ **		

Compared with BR, “↓” represents decrease in contents, “↑” represents increase in contents, * *p* < 0.05, ** *p* < 0.01.

**Table 3 molecules-23-03128-t003:** Results of *t*-test of 22 peaks from PRA showing significant difference in individual herb, herb-pair, and complicated Chinese herbal formula before and after processing (*n* = 6).

PRA			Individual Herb	Herb-Pair	Herbal Formula
No.	T_R_ (min)	Identified Compounds	*p*-Value	*p*-Value	*p*-Value
2	0.83	Sucrose	0.00678 ↓ **	0.00852 ↓ **	
4	1.97	Gallic acid	0.00250 ↓ **		
5	2.39	1-*O*-β-d-glucopyranosyl-paeonisuffrone	0.02508 ↑ *	0.04461 ↑ *	
6	3.16	6-*O*-β-d-glucopyranosyl lactinolide		0.03649 ↑ *	4.15 × 10^−5^ ↑ **
7	3.31	Mudanpioside f	0.00056 ↑ **	0.04576 ↑ *	0.00043 ↑ **
9	3.83	Oxypaeoniflora	2.79 × 10^−6^ ↑ **		0.00021 ↑ **
10	4.16	4′′-Hydroxy-3′′-methoxyalbiflorin	0.04610 ↑ *		
13	4.40	Cianidanol	0.01515 ↑ *	0.00631 ↓ **	
15	4.76	6′-*O*-β-d-glucopyranosylalbiflorin	0.02004 ↓ *		0.04757 ↓ *
18	5.13	Isomaltopaeoniflorin	1.28 × 10^−9^ ↓ **	2.58 × 10^−6^ ↓ **	
19	5.49	Albiflorin	7.64 × 10^−8^ ↑ **	0.00303 ↑ **	0.01407 ↑ *
20	5.58	Paeoniflorigenone	8.60 × 10^−10^ ↑ **	0.02864 ↑ *	0.0168 ↑ *
21	5.61	Isomaltoalbiflorin	0.00062 ↑ **	0.00952 ↑ **	0.040769 ↑ *
23	5.98	Paeoniflorin	0.04235 ↓ *		
24	6.70	Paeonol	2.80 × 10^−7^ ↑ **	0.00106 ↑ **	0.00418 ↑ **
39	10.08	Benzoic acid	5.34 × 10^−5^ ↓ **	0.00558 ↓ **	0.04072 ↓ *
41	10.18	Mudanpioside i	0.00050 ↑ **		
42	10.19	Galloylpaeoniflorin	0.00260 ↓ **		0.00856 ↑ **
48	10.68	Lactiflorin	0.03681 ↑ *		0.00508 ↓ **
62	13.96	Benzoylpaeoniflorin		0.00078 ↑ **	
63	14.09	Benzoylalbiflorin	2.40 × 10^−5^ ↑ **		0.02116 ↑ *
82	17.78	Palbinone	1.72 × 10^−6^ ↑ **		

Compared with PRA, “↓” represents decrease in contents, “↑” represents increase in contents, * *p* < 0.05, ** *p* < 0.01.

**Table 4 molecules-23-03128-t004:** Results of *t*-test of 12 peaks from AFI and GRM showing significant difference (*n* = 6).

No.	T_R_ (min)	Identified Compound	*t*-Value	*p*-Value	Source
17	4.99	Lonicerin	2.44	0.03474 ↑ *	AFI
22	5.91	Schaftoside	−4.17	0.00193 ↓ **	AFI
29	7.93	Liquiritin	8.36	8.02 × 10^−6^ ↑ **	GRM
44	10.39	Hesperetin	−4.07	0.00361 ↓ **	AFI
49	10.83	Ononin	5.62	0.00050 ↑ **	GRM
58	12.65	5,4′′-dihydroxy-3,7-dimethoxyflavone	−2.31	0.04979 ↓ *	GRM
64	14.83	Licoricesaponin A_3_	−4.69	0.00085 ↓ **	GRM
70	16.42	Licoricesaponin G_2_	−3.40	0.00677 ↓ **	GRM
72	16.55	Licoricesaponin E_2_	3.53	0.00548 ↑ **	GRM
79	17.31	Formononetin	−3.28	0.01125 ↓ *	GRM
104	20.27	Licoisoflavone a	−4.16	0.00195 ↓ **	GRM
105	20.40	Glycycoumarin	5.93	0.00014 ↑ **	GRM

Compared with SNS, “↓” represents decrease in contents, “↑” represents increase in contents, * *p* < 0.05, ** *p* < 0.01.

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
