# Peer review of "Identification and Analysis of Compound Profiles of Sinisan Based on ‘Individual Herb, Herb-Pair, Herbal Formula’ before and after Processing Using UHPLC-Q-TOF/MS Coupled with Multiple Statistical Strategy"

_molecules, 2018, doi:10.3390/molecules23123128_

Round 1
Reviewer 1 Report
This manuscript described the using UHPLC-Q-TOF/MS/MS to analyse the compound profiles of Sinisan, herb-pair of Bupleuri Radix-Paeoniae Radix and vinegar-processed Bupleuri Radix-vinegar-processed Paeoniae Radix. More than one hundred compounds were identified by MS/MS. However, in Table 2 and Table 3 only showed p-value for individual herb, herb-pair, and herbal formula. The information is not enough to know the content variation of each compound. And authors did not show the p-values were compare to what sample? In Tables 2 and 3, herb-pair and herbal formula should more clearifed as BR-PRA herb-pair and Sinisan. It is interest that some compounds can't detected in BR-PRA herb-pair and Sinisan but another compounds only present in Sinisan. Authors should discuss the vinegar-processed affect the content of compounds in decoction.
Some abbreviation such as VPBR, VPPRA in results and discussion section should be defined. In Tables 1, 2, and 3, SSq, SSg, HOSSd .... etc. should provide the full name (saikosaponin Q, sakosaponin G ? ).
In Line 83, "These components are very easy to lose an electron under mass spectrum detection, resulting in a better mass response in negative ion mode than in positive one." It should be "These components are very easy to lose a proton under mass spectrum detection, resulting in a better mass response in negative ion mode than in positive one."
The format of Figure 3 was not a good present method. Authors should change another format to more clear express the result.
In Line 175, "peak No. 89 is SSg." should be "peak No. 94 is SSg."?
Previous paper has reported after decoction the structures of saikosaponins a,c, and d will be change to saikosaponins b1, b2 and h. It should cited the related reference to support the same result. [Chinese Journal of Experimental Traditional Medical Formulae,18, 155 (2012), Journal of Beijing University of Traditional Chinese Medicine, 30, 115 (2007)]
Author Response
For dear reviewer #1:
● This manuscript described the using UHPLC-Q-TOF/MS/MS to analyse the compound profiles of Sinisan, herb-pair of Bupleuri Radix-Paeoniae Radix and vinegar-processed Bupleuri Radix-vinegar-processed Paeoniae Radix. More than one hundred compounds were identified by MS/MS. However, in Table 2 and Table 3 only showed p-value for individual herb, herb-pair, and herbal formula. The information is not enough to know the content variation of each compound. And authors did not show the p-values were compare to what sample? In Tables 2 and 3, herb-pair and herbal formula should more clearifed as BR-PRA herb-pair and Sinisan. It is interest that some compounds can't detected in BR-PRA herb-pair and Sinisan but another compounds only present in Sinisan. Authors should discuss the vinegar-processed affect the content of compounds in decoction.
Answer: Thank you very much for your criticism and suggestions. Table 2 and Table 3 actually only showed p-value for individual herb, herb-pair, and herbal formula and the content variations of each compound were shown in Figure 3. In Fact, we made a comparison in individual herb, herb-pair, and herbal formula before and after processing (The corresponding modified parts in the revised manuscript: Page 10, line 111). The reason why we didn’t clearify herb-pair and herbal formula as BR-PRA herb-pair and Sinisan was that (for example, in Table 2) individual herb included BR and VPBR, herb-pair included BR-PRA herb-pair and VPBR-VPPRA herb-pair, and herbal formula included Sinisan and Sinisan containing VPBR and VPPRA. These full names were too long for a table and affected the appearance of the tables.
Processing with vinegar and formula compatibility can both regulate the acidity and alkalinity of the solution and promote the changes in chemical composition, such as: hydrolysis reaction, isomerization reaction, etc., resulting in increased or decreased dissolution of some components (The corresponding modified parts in the revised manuscript: Page 12, lines 144-147). However the changes were complex, we took saikosaponins and paeonol as an example for detailed analysis in the revised manuscript.
●Some abbreviation such as VPBR, VPPRA in results and discussion section should be defined. In Tables 1, 2, and 3, SSq, SSg, HOSSd .... etc. should provide the full name (saikosaponin Q, sakosaponin G ? ).
Answer: Thank you very much for your kind suggestions. We have carefully checked all the full names corresponding to abbreviations throughout the paper, and the full name of saikosaponin corresponding to abbreviation of SS had been introduced in the ‘Abbreviations’ section in the first page in the revised manuscript.
●In Line 83, "These components are very easy to lose an electron under mass spectrum detection, resulting in a better mass response in negative ion mode than in positive one." It should be "These components are very easy to lose a proton under mass spectrum detection, resulting in a better mass response in negative ion mode than in positive one."
Answer: Thank you very much for your kind suggestions. The corrections had been done in the revised manuscript (The corresponding modified parts in the revised manuscript: Page 2, line 84).
●The format of Figure 3 was not a good present method. Authors should change another format to more clear express the result.
Answer: Thank you very much for your kind suggestions. The current format of Figure 3 is the more suitable and intuitive present method since the changes of each component in individual herb, herb-pair, and formula of SNS before and after processing should be stacked together for comparison. We hope the current format of Figure 3 could be retained if possible.
●In Line 175, "peak No. 89 is SSg." should be "peak No. 94 is SSg."?
Answer: Thank you very much for your kind suggestions. The corrections had been done in the revised manuscript (The corresponding modified parts in the revised manuscript: Page 13, line 178).
●Previous paper has reported after decoction the structures of saikosaponins a,c, and d will be change to saikosaponins b1, b2 and h. It should cited the related reference to support the same result. [Chinese Journal of Experimental Traditional Medical Formulae,18, 155 (2012), Journal of Beijing University of Traditional Chinese Medicine, 30, 115 (2007)]
Answer: Thank you very much for your kind suggestions and the recommended references for us to study. After carefully reading the suggested references, we learnt a lot and cited these helpful references into our revised manuscript (The corresponding modified parts in the revised manuscript: Page 17, lines 315-319).
Reviewer 2 Report
The paper by Zhou et al. deals with an important matter from a traditional Chinese medicine point of view. The study concerns identification and analysis of compound profiles of Sinisan medicinal formulae consisted of Bupleuri Radix, Paeoniae Radix Alba, Aurantii Fructus Immaturus and Glycyrrhizae Radix et Rhizoma Praeparata Cum Melle. Two controversial plant ingredients were studied in detail, i.e., Bupleuri Radix and Paeoniae Radix Alba, herb-pair of both roots, and Sinisan formulate. Using UHPLC-Q-TOF/MS 122 constituent of Bupleuri Radix, Paeoniae Radix Alba, Aurantii Fructus Immaturus and Glycyrrhizae Radix et Rhizoma Praeparata Cum Melle were identified (Table 2). Furthermore, based on three progressive levels of chemical constituent comparison (two individual herbs, herbs pair, and herbal formulae – before and after processing – data after processing is missing), the Authors suggest that processing of herbal medicine or changing medicinal formulae compatibility (data is missing) may result in changing of chemical composition and pharmaceutical efficacy – it is an obvious conclusion.
Other comments:
Title of the manuscript should be more representative and adequate to conducted work, e.g. Identification and analysis of compound profiles of Sinisan formula and their ingredients using UHPLC-Q-TOF/MS,
herb-pair of Bupleuri Radix and Paeoniae Radix Alba – data about ratio of both roots in a mixture is missing,
Table 1 – data compiled in Table 1 are original or adopted from references 13-15, as shown in line 85,
data on chemical constituents of vinegar-processed Bupleuri Radix and Paeoniae Radix Alba are missing,
the quality of figures 1 and 2 is low, also, the size of symbols in the figures axis are inappropriate,
how was the matrices for PCA calculations constructed (rows, columns), the values of total variances explained by the first two principal component are missing,
line 122 – “For BR, 22 peaks were shown ... “, but in Table 2 there are 26 peaks for BR (line 126); the same for PRA (line 129 – Table 3, 22 peaks, line 132 – 20 peaks) – these differences should be explained,
line 198 – “The decoction of BR and PRA were prepared” instead of “The decoction of BR was prepared”,
line 206 – the decoctions of VPBR and VPPRA were not prepared using the same procedures as individuals herbs, because vinegar was used,
the standard abbreviation of a journal’s name should be used in the reference section.
Author Response
For dear reviewer #2:
For dear reviewer #2:
●Title of the manuscript should be more representative and adequate to conducted work, e.g. Identification and analysis of compound profiles of Sinisan formula and their ingredients using UHPLC-Q-TOF/MS
Answer: Thank you very much for your kind suggestions. Due to the contents of our manuscript involved to ‘individual herb, herb-pair, formula’ and ‘processing’. After careful consideration, we used ‘Identification and analysis of compound profiles of Sinisan based on ‘individual herb, herb-pair, formula’ before and after processing using UHPLC-Q-TOF/MS coupled with multiple statistical strategy’ as the title of our revised manuscript (The corresponding modified parts in the revised manuscript: Page 1, lines 2-6).
●herb-pair of Bupleuri Radix and Paeoniae Radix Alba – data about ratio of both roots in a mixture is missing
Answer: Thank you very much for your kind suggestions. In the part (3.2. Sample preparation) of the manuscript, it showed that the ratio of herb-pair of Bupleuri Radix and Paeoniae Radix Alba in a mixture was 1 : 1 (4 g : 4 g).
●Table 1 – data compiled in Table 1 are original or adopted from references 13-15, as shown in line 85
Answer: Thank you very much for your kind suggestions. Data compiled in Table 1 are original. But the peak sequence of some isomers is according to references, for example, the peak sequence of SSa, SSb2, SSb1, and SSd adopted from reference 13.
●Data on chemical constituents of vinegar-processed Bupleuri Radix and Paeoniae Radix Alba are missing
Answer: Thank you very much for your kind suggestions. We are quite sorry that our misrepresentation had led your misunderstanding. Actually, data on chemical constituents of vinegar-processed Bupleuri Radix and Paeoniae Radix Alba was also in Table 1. (The corresponding modified parts in the revised paper: Page 2, line 83).
●The quality of figures 1 and 2 is low, also, the size of symbols in the figures axis are inappropriate
Answer: Thank you very much for your kind suggestions. We have improved the quality of figures 1 and 2 and also revised the size of symbols in the figures axis.
●How was the matrices for PCA calculations constructed (rows, columns), the values of total variances explained by the first two principal component are missing.
Answer: Thank you very much for your kind suggestions. Rows were constructed by m/z and retention time. Columns were constructed by 12 batches of samples. The values of total variances explained by the first two principal component were shown in the table below.
A | B | C | D | E | F | G | H | |
PC1 | 52.3% | 36.7% | 32.8% | 26.5% | 35.9% | 35.1% | 28.0% | 34.4% |
PC2 | 7.7% | 13.9% | 15.4% | 17.0% | 17.0% | 13.1% | 18.5% | 18.6% |
●Line 122 – “For BR, 22 peaks were shown ... “, but in Table 2 there are 26 peaks for BR (line 126); the same for PRA (line 129 – Table 3, 22 peaks, line 132 – 20 peaks) – these differences should be explained
Answer: Thank you very much for your kind suggestions. In Table 2 there are 26 peaks for BR in individual herb, herb-pair, and complicated Chinese herbal formula before and after processing. For individual herb (BR), 22 peaks were shown significant difference. For example, HOSSd (in Table 2) just showed significant difference in herb-pair and didn’t show significant difference in individual herb before and after processing. Also, the same explanation was for PRA.
●Line 198 – “The decoction of BR and PRA were prepared” instead of “The decoction of BR was prepared”
Answer: Thank you very much for your kind suggestions. The answer is contained in the answer to the next question (see below).
●Line 206 – the decoctions of VPBR and VPPRA were not prepared using the same procedures as individuals herbs, because vinegar was used
Answer: Thank you very much for your kind suggestions and the two questions above. According to Page 14, lines 197-198 in the revised manuscript, VPBR and VPPRA had been prepared first with vinegar according to the processing standards described in Chinese Pharmacopoeia (Edition 2015, Part Four). And then, the decoctions of VPBR and VPPRA were prepared using the same procedures as the decoctions of BR and PRA.
●The standard abbreviation of a journal’s name should be used in the reference
Section
Answer: Thank you very much for your kind suggestions. We have carefully checked the standard abbreviation throughout the reference Section, and the corrections were marked with red font in the revised manuscript.
Round 2
Reviewer 1 Report
The manuscript has been revised and it can accept in present form.
Reviewer 2 Report
I recommended the manuscript for publication in Molecules.